# Effectiveness and residual activity of four common insecticides used in the Mississippi Delta to control tarnished plant bugs in cotton

**Maribel Portilla**[1]*, **Nathan Little**[1], **Clint Allen**[1], **Yu Cheng Zhu**[2]

1 Southern Insect Management Research Unit, USDA-ARS, Stoneville, MS, United States of America,
2 Pollinator Health in Southern Crop Ecosystem Research Unit, USDA-ARS, Stoneville, MS, United States of America

* Maribel.portilla@usda.gov

**Data Availability Statement:** All relevant data are within the manuscript and its Supporting Information file.

## Abstract

The tarnished plant bug, (TPB) *Lygus lineolaris* Palisot de Beauvois (Hemiptera: Miridae) is a key pest of cotton in the midsouth region and some areas of the eastern United States. Its control methods have been solely based on chemical insecticides which has contributed to insecticidal resistance and shortened residual periods for control of this insect pest. This study was conducted over a two-year period and examined the efficacy and residual effect of four commercial insecticides including lambda-cyhalothrin (pyrethroid), acephate (organophosphate), imidacloprid (neonicotinoid), and sulfoxaflor (sulfoxamine). The effectiveness and residual effects of these insecticides were determined by application on cotton field plots on four different dates during each season using three different concentrations (high: highest labeled commercial dose (CD), medium: 1/10 of the CD, low: 1/100 of the CD) on field cotton plots. Four groups of cotton leaves were randomly pulled from each treated plot and control 0-, 2-, 4-, 7-, and 9-days post treatment (DPT) and exposed to a lab colony of TPB adults. One extra leaf sample/ plot/ spray /DPT interval (0-2-4-7-9-11) during 2016 was randomly collected from the high concentration plots and sent to Mississippi State Chemical Laboratory for residual analysis. Mortality of TPB adults was greatest for those placed on leaves sprayed with the organophosphate insecticide with mortalities (%) of 81.7±23.4 and 63.3±28.8 (SE) 1-day after exposure (DAE) on leaves 0-DPT with the high concentration for 2016 and 2017, respectively, reaching 94.5±9.5 and 95.4±7.6 6-DAE each year. Mortality to all insecticides continued until 9 and 4-DPT for high and medium concentrations, respectively. However, organophosphate (39.4±28.6) and pyrethroid (24.4±9.9) exhibited higher mortality than sulfoxamine (10.6±6.6) and the neonicotinoid (4.0±1.5) 7-DAE on 9-DPT leaves with the high concentration. Based on our results using the current assay procedure, TPB adults were significantly more susceptible to contact than systemic insecticides and due to its residual effect, organophosphate could kill over 80% of the TPB population 7-DPT.

**Funding:** The author(s) received no specific funding for this work.

**Competing interests:** The authors have declared that no competing interest exist.

## Introduction

The tarnished plant bug, *Lygus lineolaris* Palisot de Beauvois (Hemiptera: Miridae) is the most common phytophagous species of the genus *Lygus* in North America and is widely distributed from Mexico to Alaska [1, 2]. *Lygus lineolaris* is a pest of economic importance in various agronomic crops across the United States [3]. This insect became one of the most yield limiting pests of cotton, *Gossypium hirsutum* L. (Malvales: Malvaceae) in the mid-southern U.S production system following the eradication of the boll weevil, *Anthonomus grandis* Boheman (Coleoptera: Curculionidae), and the subsequent introduction and widespread adoption of the genetically modified cotton varieties to control heliothines (collectively, the bollworm, *Helicoverpa zea* (Bodie) and tobacco budworm, *Chloridea virescens* (F.)) [3, 4]. Prior to the eradication program and the introduction of transgenic cotton varieties, boll weevils and heliothines were primarily controlled with large-scale applications of organophosphates and pyrethroids (broad-spectrum insecticides). Repeated exposure to these broad-spectrum insecticides likely contributed to the development of insecticide resistance in *L. lineolaris* to several pyrethroids and some organophosphates including acephate [5, 6]. The first report of *L. lineolaris* resistance to pyrethroids (bifenthrin and permethrin) was in 1993 [7], followed closely by the organophosphate methyl parathion in 1994 [8]. Both reports were from Mississippi Delta populations of *L. lineolaris*, which preceded the introduction of transgenic Bt cotton expressing insecticidal endotoxin from the soil bacterium, *Bacillus thuringiensis* (Bt) (Berliner) in the southern U.S. in 1996 [3] the start of boll weevil eradication in 1997 and predates outbreak levels of *L. lineolaris* in cotton requiring direct control [3, 7]. Since then, the number of *L. lineolaris* populations resistant to other classes of insecticides including carbamates and neoniconinoids have increased and become widespread across the southern U.S. [9–15]. The evolution of insecticide resistance in *L. lineolaris* is a major threat not only to cotton producers but to the general agricultural and public health because these increased insecticide applications due to resistance is not sustainable economically or environmentally [3, 11, 16].

Currently, multiple applications of foliar insecticides are required to adequately control *L. lineolaris* across U.S. cotton production regions [1, 17]. During the 2022 growing season, an average of 3.4 insecticide applications were being applied to control *L. lineolaris* in the U.S., which ranged from 1 application in Texas to 5.5 in the Mississippi Delta region [18]. Despite the intensity of insecticide use during this year and for the past 30 years, *Lygus* spp. continued to devastate cotton more than any other insect in all cotton producing states, except Texas [19–23]. *Lygus* spp. infested a total of 2.1 million ha. in 2022 with total losses exceeding $ 324 million dollars [18]. Some studies suggested, that due to the inherent insecticidal resistance in *Lygus* spp., the residual period of insecticide activity has shortened, requiring tank mixing multiple insecticides with differing modes of action and / or two sequential applications with intervals of 4–7 d rather than two single applications [1, 17]. In general, cotton farmers around the word douse cotton crops in $ 2–3 billion worth of pesticides annually, of which $ 819 million has been classified as hazardous by the World Health Organization [24].

It has been well documented that *L. lineolaris* insecticide resistance varied between populations and among seasons [4, 5, 11, 25]. Snodgrass et al. [4] reported declines in pyrethroid resistance from the fall of one year to the spring of the following year. They mentioned that those declines are caused by the recessive nature of pyrethroid-resistance alleles and their dilution with susceptible alleles during mating. In addition, laboratory studies reported that levels of esterase, glutathione S-transferase (GST), and cytochrome P450 monoxygenase (P450) activity are directly correlated with resistance intensity to any class of insecticide used to control *L. lineolaris* [26–33]. Several authors reported elevated levels of esterase, GST, and P450 in populations of *L. lineolaris* from the Mississippi Delta and matched the highest level to the

intensity of pesticide use in certain production areas [4, 12, 16]. However, a recent study demonstrated that *L. lineolaris* lost its resistance to five pyrethroids and two neonicotinoids after 36 consecutive generation under laboratory conditions without insecticide exposure. Authors mentioned that the colony lowered their activity of esterases, GST, and P450 to levels similar to the susceptible colony [34], meaning, that the insecticide resistance of *L. lineolaris* can fade away or diminishes in the absence of selection pressure. All these explained why *L. lineolaris* populations should be managed differently across seasons and various cotton production areas. They also stressed that resistance monitoring, insecticide mode of action selection, and residual effects would be vital to managing *L. lineolaris* populations as part of an effective integrated management strategy.

Today, regardless of inherent resistance levels, organophosphates and pyrethroids are the most common contact insecticides used for *L. lineolaris* control [3]. However, during the last decade newer insecticides have been introduced, including neonicotinoids and sufolxamines [2]. Therefore, this study was conducted to evaluate the efficacy and residual activity of four potential insecticides (one per class) used in Mississippi Delta. The results for this study will provide a better understanding of selecting insecticide used based in residual activity and mode of action to control *L. lineolaris* as an integrated pest management.

## Materials and methods

### Field plots and applications

The efficacy and residual activity of two contact insecticides and two systemic insecticides on *L. lineolaris* were evaluated in a non-Bt cotton cultivar in 2016 and 2017 on the Southern Insect Management Research Unit's (SIMRU) research farm near Stoneville, MS (latitude 33.3456, longitude -90.9168). The experiment was set up with plots within a randomized complete block designed with four replications (4 blocks) (16 plots/block). Each plot consisted of eight rows of non-Bt cotton (DP1441RF®, Delta and Pine Land Company ™, Scott, MS) with 101.6 cm wide rows approximately 100 m long. Ad hoc applications of herbicides and plant growth regulator (mepiquat chloride, Loveland Products, Inc., Morgantown, KY) were applied equally to all plots within a given year of the study. Each of the following treatments were randomly assigned to each cotton plot within each block of the experiment: 1) acephate (organophosphate) (Bracket 90 WSP™, AgriSolutions, Winfield Solutions, LLC, St. Paul, MN, USA), 2) lambda-cyhalothrin (pyrethroid) (Karate™, Syngenta Crop Protection, Inc. Greenboro, NC, USA), 3) imidacloprid (neonicotinoid) (Admire Pro™, Bayer CropScience LP, Research Triangle Park, USA), 4) sulfoxaflor (sulfoxamine) (Transform WG™, Corteva Agriscience, Indianapolis, IN, USA), and 5) untreated control. Each insecticide was applied at three different concentrations (treatments) as follow: high: highest labeled commercial dose (CD) for *L. lineolaris* in cotton, medium: 1/10 of the CD, and low: 1/100 of the CD. Each block (replication) was sprayed on four different dates within a given year of the study targeting each spray approximately two weeks apart. From the eight cotton rows/plot only the middle four rows were treated. All treatments were applied with a Lee Avenger sprayer (LeeAgra, Inc. Lubbock, TX) equipped with a ten multi-boom spray system (BellSpray Inc., Opelousa, LA) with hollow cone nozzle (TX-VS8, Tee Jet Technologies, Glendale Height, IL), which was calibrated to deliver 93.54 L of spray solution per ha. Application dates, contact and systemic insecticides used, and associated rates are listed in Table 1. Three-four- h after spray (0-day post treatment) (0-DPT), and 2, 4, 7, and 9-DPT thereafter, four groups (subsamples) of 15 leaves / group from the cotton top nodes were randomly collected from each plot (720 leaves/ sprayed block/ evaluation day) and taken to the lab for bioassays. Leaf samples were pulled from plots from the lowest to the highest concentration to avoid cross contamination.

**Table 1. Insecticides and treatment rates for the efficacy and residual activity on *L. lineolaris* in cotton leaves during 2016\* and 2017\*\*.**

| Insecticide | Rate | Dose/acre | Dose/hectare |
|---|---|---|---|
| Acephate | Low | 4.55 g/a | 11.2 g/ha |
| | Medium | 45.5 g/a | 112.4 g/ha |
| | High | 455 g/a | 1124 |
| Lambda-cyhalothrin | Low | 0.75 mL/a | 1.9 mL/ha |
| | Medium | 7.5 mL/a | 18.5mL/ha |
| | High | 75 mL/a | 185.3 mL/ha |
| Imidacloprid | Low | 0.5 mL/a | 1.2 mL/ha |
| | Medium | 5 mL/a | 12.4 mL/ha |
| | High | 50 mL/a | 123.6 mL/ha |
| Sulfoxaflor | Low | 0.65 g/a | 1.6 g/ha |
| | Medium | 6.5 g/a | 16.1 g/ha |
| | High | 65 g/a | 160.6 g/ha |

Application dates:

\*2016: July 12, July 20, July 25, and August 8

\*\*2017: July 17, July 30, August 21, and Sep 6 (third and fourth spray were applied when the weather permitting)

## Leaf samples for insecticide residual analysis test

From at least two sprays, each insecticide had leaf samples pulled at 0- day post treatment (0-DPT), 2, 4, 7, 9, and 11-DPT. The top of 10 cotton plants (high concentration plots) were marked in each plot with flagging tape just before the plots were sprayed. One leaf from each flagged plant was pulled (10 leaves/sample/plot/insecticide) at each evaluation time. Leaf samples were sent to the Mississippi State University Chemistry Laboratory for leaves that were present at the time of the insecticide spray. The insecticide analysis test was done for 2016 only.

## Insect colonies

A laboratory-reared *L. lineolaris* colony has been maintained at the United States Department of Agriculture Agricultural Research Service (USDA ARS), Southern Insect Management Research Unit (SIMRU) in Stoneville, MS since 2011 [35], but was stablished in 1998 at the USDA-ARS Biological Control Rearing and Research Unit in Starkville, MS [36]. This laboratory colony is routinely reared following procedures outlined by Portilla et al. [36], which was designed for mass production of even-aged individuals. Insects were held in environmental chambers with a photoperiod of 12:12 (L: D) h, 27C, and 60% RH. Mixed-sex adults 2-d old were used for this study. This insect colony has not been exposed to insecticides and/or have had field population infusions, which make the insects particularly valuable for screening insecticides.

## Bioassay procedure

Laboratory essays were conducted to determine the efficacy and residual activity of the insecticide treatments. Four groups of 15 leaves randomly collected from each plot at 0, 2, 4, 7, 9 (DPT) (60 leaves / plot) (960 leaves / block / evaluation time) (4,800 leaves-insects/spray-replicate) were taken to the lab and placed individually into 30-mL plastic cups (T-125 SOLO-cup, Pleasant Prairie, WI). A 2-d old *L. lineolaris* adult (unknown sex) was released to each cup that contained the treated leaf. The lids for the cups had three holes for ventilation (3 mm

diameter). Adults were examined daily (7-d period) for mortality. This process was repeated for each evaluation time (0, 2, 4, 7, and 9-DPT) and for each spray (replication).

## Analyses

All experiments were analyzed using SAS 9.4 (SAS Institute 213) [37]. A randomized complete block design with factorial arrangements (treatment concentration-days of exposure x sub-samples x spray) (32 x 4 x 4) per each evaluation time was used for cumulative mortality. Mortality percentage and residual analysis test were analyzed by using ANOVA followed by Tukey's HSD. Probit analyses were used to develop regressions for estimating lethal concentrations ($LC_{50}$) of each tested insecticide (SAS Institute 2013). Percent mortalities for each treatment were corrected for control effects using Abbott's formula [38]. Resistance ratios ($RR_{50}$) and 95% CI were calculated using Robertson and Priestler's formula [39]. Bioassays were considered significant when the slope of the line was significant ($P < 0.05$).

## Results

### Effectivity and residual activity of acephate by leaf sample bioassay

There were statistically significant differences in cumulative mortality of acephate (organophosphate) to *L. lineolaris* among concentrations and day of exposure at each evaluation of DPT (Table 2). The organophosphate had the greatest effect on TPB adults with mortality (%) of 48.75 ± 18.3 0-day after exposure (DAE) on leaves 0-DPT and its residual effect continued in leaves pulled 9-DAT. The high residual activity was sustained until 4-DPT period with mortality over 90% on 7-DAE for 2016 (Fig 1A–1C), decreasing then to 80 ± 28.8 on 7-DAT (Fig 1D) and 39 ± 8.27 on 9-DAT (Fig 1E). A similar cumulative mortality trend was observed only for the high concentration in 2017 with no significant differences on cumulative mortality from 4 to 7-DAE for both years (Fig 1A and 1F). The high concentration had a better control in 2016, while a better dose response was observed in 2017 with a greater residual activity for the lower concentrations (Fig 1F–1I). Heavy precipitation affected residual activity in 2017 for the 7-DPT (Fig 1I), lowering its residual effect 5.14-fold (12.08 ± 2.93) compared to the cumulative mortality at 4-DPT (62.08 ± 34.74) or 15-fold lower if compared with the 7-DPT evaluation in 2016. Due to its low residual activity 7-DAT and because the weather, no leaf samples were collected for the 9-DPT evaluation in 2017 for the fourth spray (replicate). Less than 2% cumulative mortality was observed in control with no significant differences among medium and low concentrations in 2016 for 4, 7, and 9-DPT (Fig 1C–1E) and in 2017 for 7-DPT only (Fig 1I). The low residual activity observed in 2017, could have caused by the precipitation. The rainfall obtained during the second spray (July 30, 2017) was: 0.4 in (2-DPT), 0.2 in (4-DPT), 0.13 (7-DPT), and 4.22 in (9-DPT); for the third spray (August 21, 2017) was: 0.44 in (7-DPT) and 1.7 in (9-DPT); the precipitation for the fourth spray (September 6, 2017) was: 0.15 in (7-DPT) and 0.76 in (9-DPT). Thus, due to the low residual activity, no data was plotted for day 9 in 2017. Table 3 shows that the probit model produced a good fit of the data for the $LC_{50}$ estimated values for both years. The residual activity is measured by the estimation of the $LC_{50}$s and resistance ratios ($RR50$s). The lethal concentration and $RR_{50}$s for 2016 and 2017 increased significantly over the time (DPT). Therefore, the lower the residual activity the higher $LC_{50}$s and $RR_{50}$s.

### Effectivity and residual activity of lambda-cyhalothrin by leaf sample bioassay

As observed with acephate, there were statistically significant differences in cumulative mortality of lambda-cyhalothrin (pyrethroid) to *L. lineolaris* among concentrations and day of

**Table 2. Overall general lineal model for cumulative mortality (7-d period) of insecticides and treatments rates on *L. lineolaris* in cotton leaves at different days post treatment during 2016 and 2017.**

| Insecticide | Days Post Treatment (DPT) | | | | |
|---|---|---|---|---|---|
| | 0-D | 2-D | 4-D | 7-D | 9-D |
| 2016 | | | | | |
| Acephate | F = 214.2 $P = < .0001$ | F = 239.45 $P = < .0001$ | F = 159.25 $P = < .0001$ | F = 8.67 $P = < .0001$ | F = 14.39 $P = < .0001$ |
| Lambda-cyhalothrin | F = 21.93 $P = < .0001$ | F = 90.19 $P = < .0001$ | F = 67.60 $P = < .0001$ | F = 16.57 $P = < .0001$ | F = 21.16 $P = < .0001$ |
| Imidacloprid | F = 23.79 $P = < .0001$ | F = 7.70 $P = < .0001$ | F = 9.95 $P = < .0001$ | F = 13.18 $P = < .0001$ | F = 4.67 $P = < .0001$ |
| Sulfoxaflor | F = 78.69 $P = < .0001$ | F = 38.09 $P = < .0001$ | F = 32.01 $P = < .0001$ | F = 8.67 $P = < .0001$ | F = 10.37 $P = < .0001$ |
| 2017 | | | | | |
| Acephate | F = 107.09 $P = < .0001$ | F = 79.76 $P = < .0001$ | F = 26.20 $P = < .0001$ | F = 10.94 $P = < .0001$ | |
| Lambda-cyhalothrin | F = 45.97 $P = < .0001$ | F = 45.24 $P = < .0001$ | F = 24.33 $P = < .0001$ | F = 11.99 $P = < .0001$ | |
| Imidacloprid | F = 14.06 $P = < .0001$ | F = 8.69 $P = < .0001$ | F = 7.07 $P = < .0001$ | F = 5.04 $P = < .0001$ | |
| Sulfoxaflor | F = 79.76 $P = < .0001$ | F = 7.89 $P = < .0001$ | F = 7.49 $P = < .0001$ | F = 6.87 $P = < .0001$ | |

n = 60

DF = 31, 3 for all treatments at 0, 2, and 4 DPT 2016

DF = 31, 2 for all treatments at 7 and 9 DPT 2017

DF = 31, 3 for all treatments at 0, 2, 4, and 7 DPT 2017

exposure at each evaluation period of DPT (Table 4). However, the pyrethroid had lower effect relative to the organophosphate on TPB adults. Although its residual effect continued to 9-DPT for the high and medium concentration, its initial mortality on 0-DAE did not exceed 10% for 2016 (8 ± 1.6) but increased over 20% in 2017 (22.50 ± 6.85) on leaves pulled on 0-DPT for the high concentration (Fig 2A and 2F). Fig 2A and 2F show that initial mortality for the medium concentration started 1-DAE, while for the low concentration, started 5-DAE for both years on leaves pulled on 0-DPT. Cumulative mortality for the high concentration slowly reached about 80% 7-DAE (Fig 2A and 2F). This residual activity continued in leaves pulled 2-DPT (79.17 ± 13.30) (Fig 2B) and 4-DPT (69.58 ± 22.43) (Fig 2C) for 2016, and 2-DPT (75.0 ± 23.28) for 2017 (Fig 2G). Cumulative mortality for the medium concentration reached 35.41 ± 8.62 and 55.0 ± 7.56 for 2016 and 2017, respectively 7-DAE on leaves 0-DPT (Fig 2A and 2F). Its residual activity dropped 2.9- (12.50 ± 2.34) and 2.8-fold (20.83 ± 6.71) in 2016 and 2017, respectively on 2-DPT, lowering to less than 5.0 and 2.0% on 4-DPT for 2016 and 2017, respectively (Fig 2B and 2G). No residual activity was observed for the low concentration after 2-DPT for either year (Fig 2C and 2H). Similar mortality trends were observed for both years during the evaluation periods of 0, 2, and 4-DPT. However, similar to that observed for acephate, the residual activity of lambda-cyhalothrin was likely lessened due to the high precipitation event before the 7-DPT on 2017 (Fig 2I). Due to its low residual activity, no leaf samples were collected for the 9-DPT evaluation in 2017. Less than 2% cumulative mortality was observed in the control with no significant differences among low concentrations in 2016 for 4, 7, and 9-DPT (Fig 2C–2E) and in 2017 for 4, and 7-DPT (Fig 1H and 1I). Table 3 shows a dose-dependent effect at all evaluation period (DPT) for both years. The $LC_{50}$ value for leaves pulled 0-DPT was 33.85 mL / ha, lowering its residual activity 10.59-fold after 9 days (358.79.2

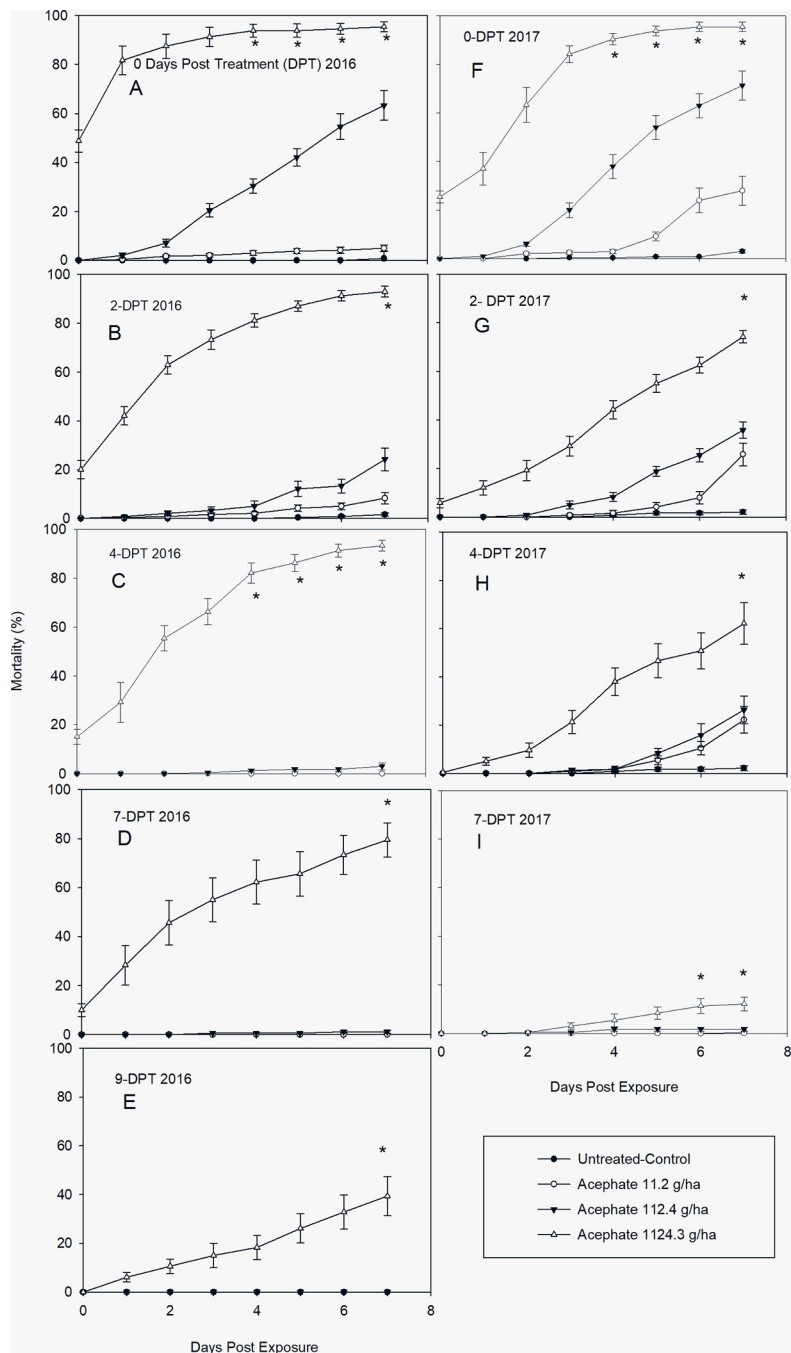

**Fig 1. Post treatment mean ± SD cumulative mortality of a laboratory-reared *L. lineolaris* colony exposed to acephate on cotton leaves sprayed in the field and collected days posttreatment (DPT).** A-E: Mortality (%) in 2016 after DPT period and exposed for 7 days. F-I: Mortality (%) in 2017 after DPT periods and exposed for 7 days (Tukey' HSD test, *p* = 0.05).

mL / ha) for 2016. In 2017 the $LC_{50}$ for 0-DPT was 15.1 mL / ha decreasing the residual activity 17.65-folds (254.1 mL / ha) after 7 days. Probit model produced a good fit of the data for the $LC_{50}$ estimated values for both years.

**Table 3. Lethal mortality response (LC$_{50}$) of *L. lineolaris* exposed to different concentrations of acephate estimated at different days post treatment on cotton leaves during 2016 and 2017.**

| Days Post Treatment | Concentration response (g / ha) | | | | | | | |
|---|---|---|---|---|---|---|---|---|
| | n | Slope ± SE | LC$_{50}$ (95%CI) | Probit Trend | | | | RR$_{50}$ (95%CI) |
| | | | | Test for Slope | | Test for GoF | | |
| | | | | $X^2$ | $P > X^2$ | $X^2$ | $P > X^2$ | |
| Year 2016 | | | | | | | | |
| 0-D | 960 | 0.743 ± 0.076 | 86.12 (63.18–115.1) | 96.320 | < .0001 | 2.8432 | < .0001 | 1.0 |
| 2-D | 960 | 0.980 ± 0.122 | 256.71 (192–342.20) | 64.592 | < .0001 | 2.6776 | < .0001 | 2.86 (1.9–4.29) |
| 4-D | 960 | 1.292 ± 0.108 | 283.18 (230.1–346.44) | 142.501 | < .0001 | 1.8312 | 0.0005 | 3.36 (2.33–4.78) |
| 7-D | 720 | 1.133 ± 0.153 | 414.15 (289.06–568.1) | 54.310 | < .0001 | 3.4325 | < .0001 | 4.84 (3.13–7.49) |
| 9-D | 720 | 0.591 ± 0.102 | 839.86 (525.6–1627.2) | 33.781 | < .0001 | 3.0014 | < .0001 | 9.87 (5.45–17–84) |
| Year 2017 | | | | | | | | |
| 0-D | 960 | 0.511 ± 0.067 | 40.06 (23.7–62.34) | 58.461 | < .0001 | 3.4623 | < .0001 | 1.0 |
| 2-D | 960 | 0.290 ± 0.039 | 200.65 (120.8–349.65) | 56.267 | < .0001 | 1.8675 | 0.0003 | 4.89 (2.46–9.74) |
| 4-D | 960 | 0.252 ± 0.689 | 582.92 (205.5–4502.3) | 13.531 | 0.0002 | 5.5291 | < .0001 | 13.99 (3.92–49.96) |
| 7-D | 960 | 0.464 ± 0.1001 | 14615 (5621–145339) | 20.903 | < .0001 | 1.1410 | 0.2349 | 391.65 (92.54–1657.44) |

LC$_{50}$ values were calculated in g / ha

Mortality was scored at 7 Days After Exposure

Differences among RR$_{50}$ values are significant if 95% CI do not 1nclude 1.0

RR$_{50}$ and 95% CI calculated using formula from Robertson et al. [39]

## Effectivity and residual activity of imidacloprid by leaf sample bioassay

The effectiveness and residual activity of the neonicotinoid imidacloprid, is displayed in Fig 3. There were statistically significant differences in cumulative mortality of imidacloprid to *L.*

**Table 4. Lethal mortality response (LC$_{50}$) of *L. lineolaris* exposed to different concentrations of lambda-cyhalothrin estimated at different days post treatment in cotton leaves during 2016 and 2017.**

| Days Post Treatment | Concentration response (mL / ha) | | | | | | | |
|---|---|---|---|---|---|---|---|---|
| | n | Slope ± SE | LC$_{50}$ (95%CI) | Probit Trend | | | | RR$_{50}$ (95%CI) |
| | | | | Test for Slope | | Test for GoF | | |
| | | | | $X^2$ | $P > X^2$ | $X^2$ | $P > X^2$ | |
| Year 2016 | | | | | | | | |
| 0-D | 960 | 0.340 ± 0.054 | 33.85 (14.21–77.34) | 38.895 | < .0001 | 3.3980 | < .0001 | 1 |
| 2-D | 960 | 0.895 ± 0.084 | 61.65 (42.28–94.15) | 112.569 | < .0001 | 1.4243 | 0.0307 | 1.81 (0.75–4.41) |
| 4-D | 960 | 0.539 ± 0.058 | 76.58 (61.75–93.9) | 85.057 | < .0001 | 2.3410 | < .0001 | 2.17 (0.95–4.95) |
| 7-D | 720 | 0.501 ± 0.081 | 126.52 (72–271.57) | 37.802 | < .0001 | 2.9890 | < .0001 | 3.73 (1.36–10.18) |
| 9-D | 720 | 0.406 ± 0.049 | 358.8 (212.76–742.2) | 68.614 | < .0001 | 1.2139 | 0.1826 | 10.59 (3.89–28.81) |
| Year 2017 | | | | | | | | |
| 0-D | 960 | 2.984 ± 9711 | 15.07 (-) | 0.0000 | 0.9998 | 1.2604 | 0.2467 | 1 |
| 2-D | 960 | 0.467 ± 0.552 | 36.42 (-) | 0.715 | 0.3979 | 1.7594 | 0.0136 | 1.3 (-) |
| 4-D | 960 | 0.550 ± 1.115 | 71.96 (35.68–201.1) | 24.301 | < .0001 | 0.7068 | 0.9051 | 5.22 (-) |
| 7-D | 960 | 0.750 ± 0.289 | 254.83 (147.9–1662) | 6.708 | 0.0096 | 0.8077 | 0.8177 | 17.65 (-) |

LC$_{50}$ values were calculated in mL / ha

Mortality was scored at 7 Days After Exposure

Differences among RR$_{50}$ values are significant if 95% CI do not 1nclude 1.0

RR50 and 95% CI calculated using formula from Robertson et al. [39]

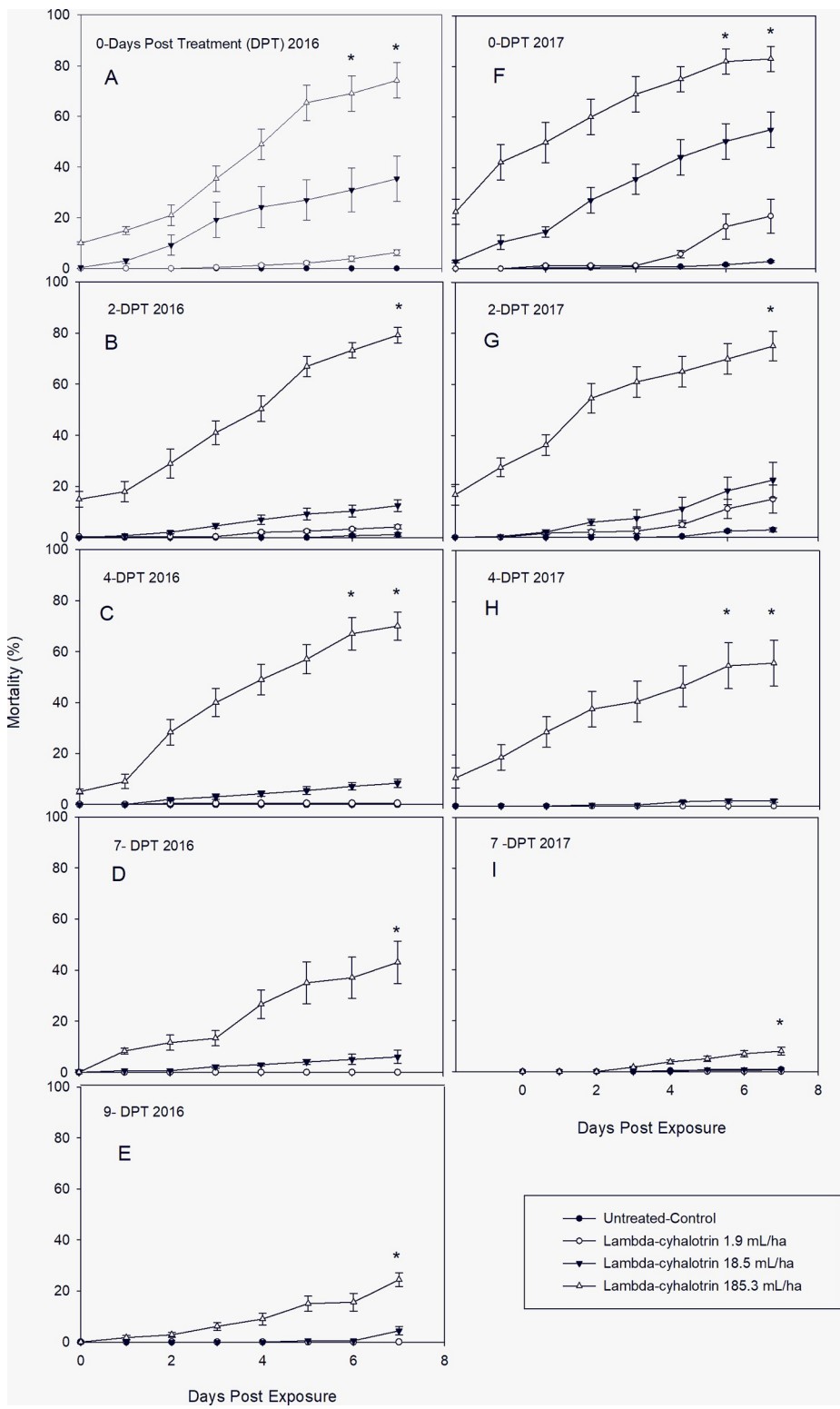

**Fig 2. Post treatment mean ± SD cumulative mortality of laboratory-reared *L. lineolaris* colony exposed to the pyrethroid insecticide lambda-cyhalothrin on cotton leaves sprayed in the field and collected days posttreatment (DPT).** A-E: Mortality (%) in 2016 after DPT periods and exposed for 7 days. F-I: Mortality (%) in 2017 after DPT periods and exposed for 7 days (Tukey' HSD test, $p = 0.05$).

*lineolaris* among concentrations and day of exposure at each evaluation period of DPT (Table 2). The efficacy of the neonicotinoid against *L. lineolaris* was distinctly inferior over organophosphate and pyrethroid in terms of mortality percentage. Although the residual effect continued to 9-DPT for the high concentration, the cumulative mortality (7-DAE) barely exceeded 30% for both years (33.33 ± 12.17 and 29.58 ± 19,77 for 2016 and 2017, respectively) on leaves pulled 0-DPT. The cumulative mortality decreased > 4-fold for 2016 (8.33 ± 3.07) but it was sustained at > 20% for 2017 (22.08 ± 7.7) on leaves pulled 2-DPT (Fig 3B and 3G). After that, the residual effect was not enough to kill more than 20% of the population in 2016 until 7-DPT period (Fig 3B–3D) and until 4-DPT period for 2017 (Fig 3G and 3H). The initial mortality of TPB adults with the highest concentration was observed 1 and 2-DAE on leaves pulled 0-DPT for 2016 and 2017, respectively. The initial mortality for the same treatment was exhibited 2 and 5-DAE for 2016 and 2017, respectively on leaves pulled 2-DPT period (Fig 3B and 3G). No differences in mortality were observed between control, medium, and low concentrations at 4, 7, and 9-DPT for 2016 (Fig 3C–3E) and 4 and 7-DPT for 2017 (Fig 3H and 3I). Mortality in untreated leaves did not occur until 6–7 DAE. Although the efficacy and residual activity was ineffective for the beginning of the treatment applications, it was very clear how the high precipitation obtained in 2017 also affected this insecticide (Fig 3I). Table 5 shows a dose-dependent effect at all evaluation periods (DPT) for 2016. The probit model did not produce a good fit for all data regarding $LC_{50}$ estimated values for 2017. The $LC_{50}$ value for leaves pulled 0-DPT in 2016 was 421.6 mL / ha, that was 3.4-fold higher that the commercial dose (123.6 mL / ha) decreasing the residual activity 11.71-fold (5505.5 mL / ha) after 9-DPT. An unexpected irregularity of $LC_{50}$ estimation was observed for 2017; therefore, the residual activity could not be measured by the estimation of $LC_{50}$s and $RR_{50}$s for this year. However, Fig 5 clearly demonstrated that the efficacy gradually decreased throughout the DPT periods.

## Effectivity and residual activity of sulfoxaflor by leaf sample bioassay

The effectiveness and residual activity of the sulfoxaflor (sulfoxamine) is presented in Fig 3. There were statistically significant differences in cumulative mortality of sulfolxaflor to *L. lineolaris* among concentrations and day of exposure at each evaluation period of DPT (Table 2). Like all insecticides tested, the residual effect of the high concentration continued until after 9-DPT for 2016 (Fig 4E) and 7-DPT for 2017 (Fig 4I). The mortality trend for sulfoxaflor is comparable to pyrethroid for the leaves pulled 0-DPT, where mortality increased gradually and reached the maximum cumulative mortality 7-DAE, for the high concentration for both years (67.91 ± 15.62 and 74.16 ± 10.0 for 2016, 2017, respectively) (Fig 4A and 4F). However, after the 2-DPT the residual effect decreased almost as fast as the neonicotinoid either in 2016 or 2017 (Fig 4G–4I). The initial mortality for the high concentration was observed 2-DAE in 2016, while in 2017 a low percentage (5.83 ± 2.42) was observed at 0-DPT. No differences in mortality were observed between control, medium, and low concentrations at 2, 4, 7, and 9-DPT for 2016 (Fig 4B–4E) and 4 and 7-DPT for 2017 (Fig 3H and 3I). Mortality in untreated leaves did not occur until 6–7 DAE. Similarly, as the rest of the insecticides, sulfoxaflor was affected by the precipitation in 2017. Table 6 shows a dose-dependent effect at all evaluation period (DPT) for 2016 and 2017. Probit model did produce a good fit for all data for the $LC_{50}$ estimated values for both years. The $LC_{50}$ value for leaves pulled 0-DPT was 91.23 g / ha, lowering its residual activity 20.48-fold after 9 days (1763.3 g / ha) for 2016. In 2017 the $LC_{50}$ for 0-DPT was 156.96 g / ha decreasing the residual activity 434.53-folds (254.1 g / ha) after 7 days. Probit model produced a good fit of the data for the $LC_{50}$ estimated values for both years except for the 2-DPT in 2017 which was higher than expected.

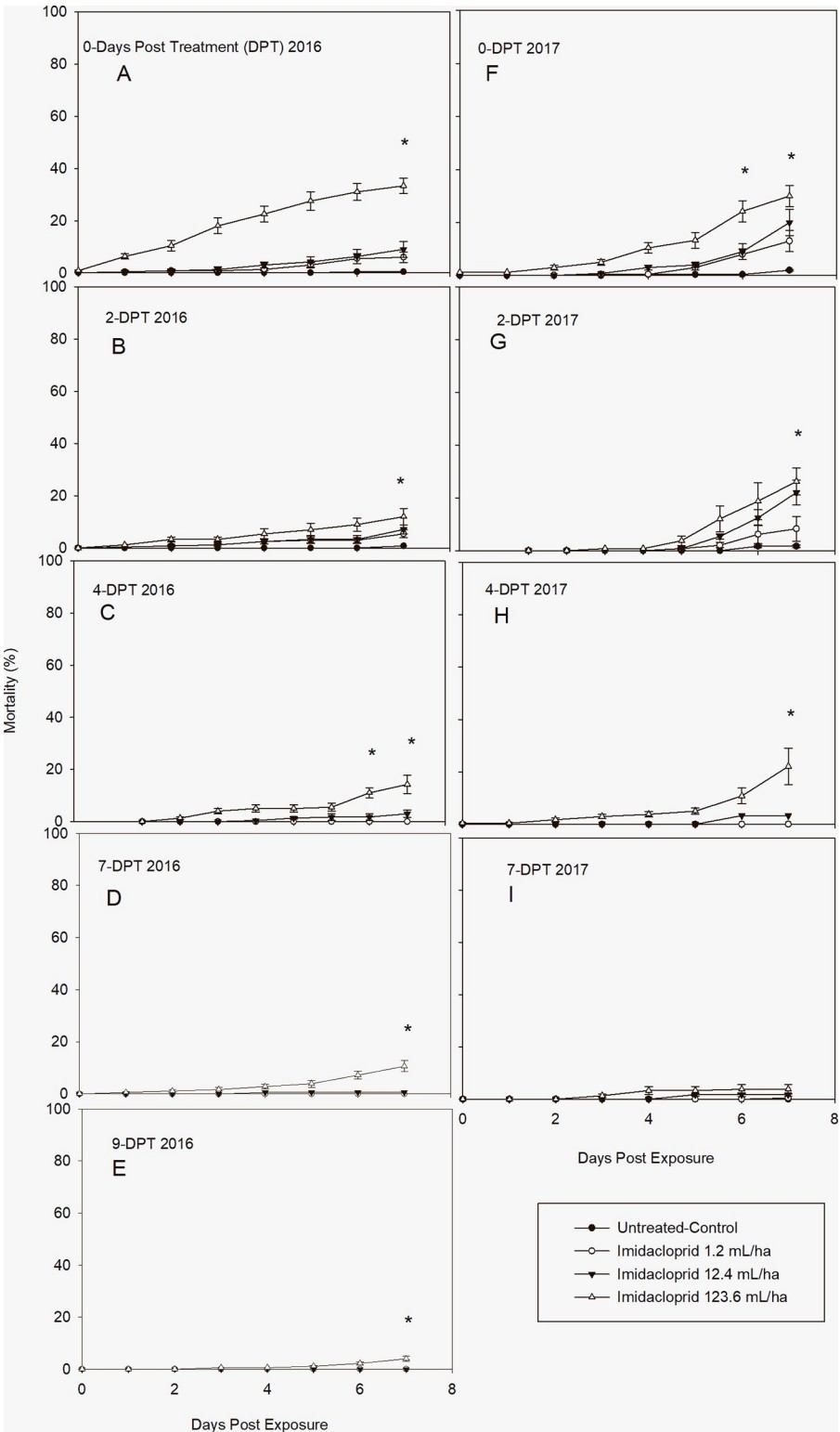

**Fig 3. Post treatment mean ± SD cumulative mortality of a laboratory-reared *L. lineolaris* colony exposed to the insecticide neonicotinoid, imidacloprid, on cotton leaves sprayed in the field and collected days posttreatment (DPT).** A-E: Mortality (%) in 2016 after DPT periods and exposed for 7 days. F-I: Mortality (%) in 2017 after DPT periods and exposed for 7 days (Tukey' HSD test, *p* = 0.05).

**Table 5. Lethal mortality response (LC$_{50}$) of *L. lineolaris* exposed to different concentrations of imidacloprid estimated at different days post treatment in cotton leaves during 2016 and 2017.**

| Days Post Treatment | Concentration response (g–mL/Ha) | | | | | | | |
|---|---|---|---|---|---|---|---|---|
| | n | Slope ± SE | LC$_{50}$ (95%CI) | Probit Trend | | | | RR$_{50}$ (95%CI) |
| | | | | Test for Slope | | Test for GoF | | |
| | | | | $X^2$ | $P > X^2$ | $X^2$ | $P > X^2$ | |
| Year 2016 | | | | | | | | |
| 0-D | 960 | 0.263 ± 0.066 | 421.6 (216.5–1215.2) | 15.910 | < .0001 | 1.8262 | 0.0005 | 1 |
| 2-D | 960 | 0.604 ± 0.268 | 456.7 (224.6–41175016) | 5.083 | 0.0242 | 1.5543 | 0.0094 | 1.09 (0.20–5.89) |
| 4-D | 960 | 0.368 ± 0.051 | 470.5 (184.6–3301) | 53.332 | < .0001 | 1.3355 | 0.0636 | 1.11 (0.27–4.56) |
| 7-D | 720 | 0.328 ± 0.061 | 798.9 (289.8–6311) | 28.740 | < .0001 | 1.4018 | 0.0601 | 1.69 (0.29–9.82) |
| 9-D | 720 | 0.255 ± 0.055 | 5505.5 (1058.6–283783) | 21.086 | < .0001 | 1.1363 | 0.2682 | 11.71 (0.85–161.18) |
| Year 2017 | | | | | | | | |
| 0-D | 960 | 0.135 ± 0.063 | 9350 (365.9–26788) | 4.702 | 0.0301 | 3.8583 | < .0001 | 34.92 (0.10–11726.55) |
| 2-D | 960 | 0.029 ± 0.084 | 6.63472E12 (-) | 0.119 | 0.7294 | 8.2031 | < .0001 | (-) |
| 4-D | 960 | 0.425 ± 0.082 | 219.9 (105.3–832.5) | 26.770 | < .0001 | 3.5297 | < .0001 | 1 |
| 7-D | 960 | 0.252 ± 0.114 | 145441 (2713–6.74672E37) | 4.921 | 0.0265 | 1.3444 | 0.0593 | 18.69 (0.51–843292) |

LC$_{50}$ values were calculated in g/Ha

Mortality was scored at 7 Days After Exposure

Differences among RR$_{50}$ values are significant if 95% CI do not 1nclude 1.0

RR50 and 95% CI calculated using formula from Robertson et al. [39]

### Leaf samples for insecticide residual analysis test

Timing of leaf samples and the residual analysis test is presented in Fig 5. Residual activity of all insecticides was detected up to 11-DPT. There were statistically significant differences in residual activity (ppm) among insecticides and period of evaluations 0, 2, 4, and 7-DPT. Acephate exhibited the greatest residual effect with the highest ppm values. It was highly significantly different among lambda cyhalothrin, imidacloprid, and sulfoxaflor on leaves collected 0-DPT: $F_{(3, 3)} = 10.22$, $P = 0.0060$; 2-DPT: $F_{(3, 3)} = 12.31$, $P = 0.0057$; 4-DPT: $F_{(3, 3)} = 8.15$, $P = 0.0593$; 7-DPT: $F_{(3, 3)} = 21.33$, $P = 0.0159$; and 9-DPT: $F_{(3, 3)} = 21.33$, $P = 0.0357$. No significant differences were observed between lambda cyhalothrin, imidacloprid, and sulfoxaflor at any of those evaluation times. No significant differences of residual activity (ppm) were found among insecticides for 11-DPT ($P = 0.5692$).

## Discussion

Environmental conditions could play an important role on insecticide residual activity. It is important to clarify that no rainfall and temperature was recorded for this investigation; yet, due to the high precipitation obtain in 2017, rainfall information was pulled from Delta Agricultural Weather Center [40] for the spray days and leaf collection DPT (See material and methods). Based in our results all insecticides tested in 2017 could have an influence persistence and degradation of foliar insecticides residues. Rainfall, especially when it occurs shortly after application, as it was observed in this investigation, can lead to immediate reduction of insecticide residues [41, 42]. This could explain the differences in mortality among years mainly for high concentrations.

Thorough knowledge of the impacts related to contact toxicity of insecticidal residues is necessary for successful integrated pest management (IPM). Contact toxicity of residues is composed of two components including initial mortality experienced soon after application

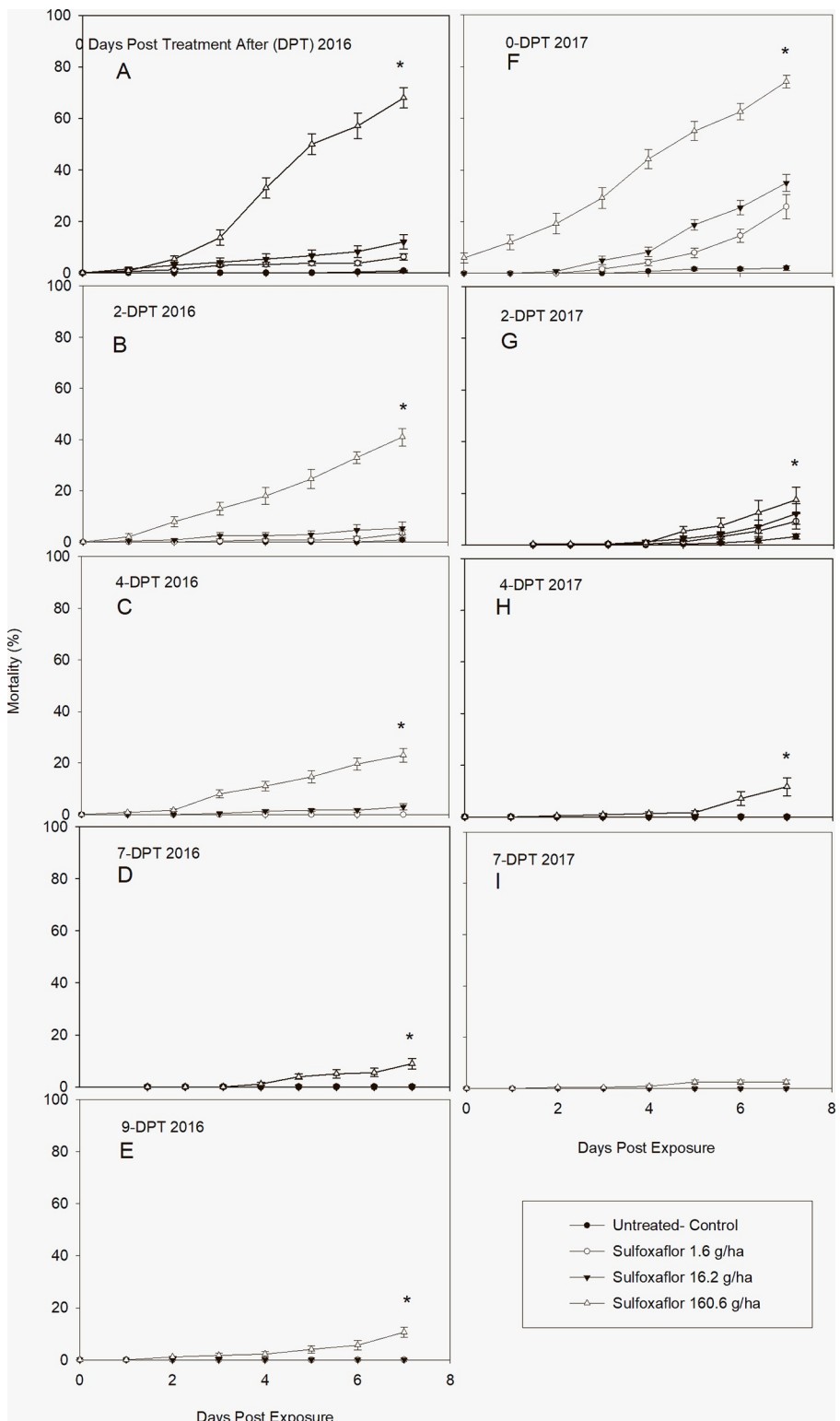

**Fig 4. Post treatment mean ± SD cumulative mortality of a laboratory-reared _L. lineolaris_ colony exposed to the sulfoxamine insecticide sulfoxaflor, on cotton leaves sprayed in the field and collected days posttreatment (DPT).** A-E: Mortality (%) in 2016 after DPT periods and exposed for 7 days. F-I: Mortality (%) in 2017 after DPT periods and exposed for 7 days (Tukey' HSD test, _p_ = 0.05).

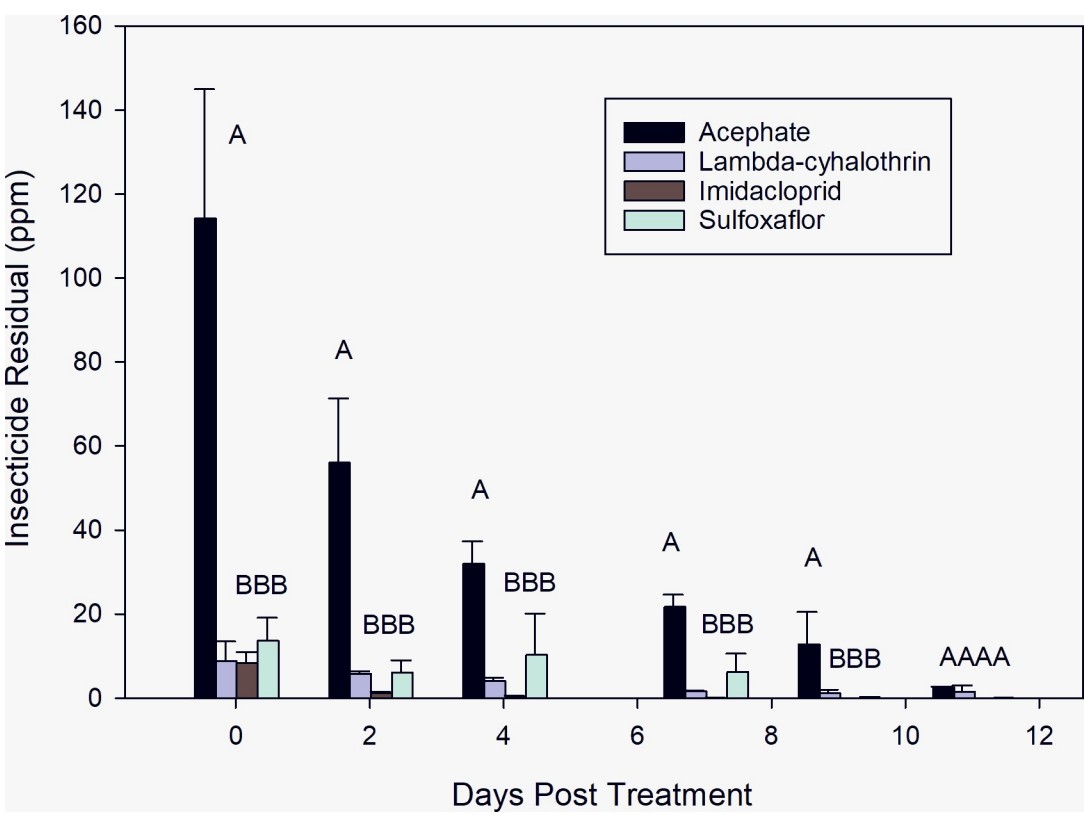

**Fig 5. Insecticidal residual test of four classes of insecticides on leaf samples pulled at 0, 2, 4-, 7-, 9-, and 11-days post treatment periods.** (Tukey' HSD test, $p$ = 0.05).

**Table 6. Lethal mortality response (LC$_{50}$) of *L. lineolaris* exposed to different concentrations of sulfoxaflor estimated at different days post treatment in cotton leaves during 2016 and 2017.**

| Days Post Treatment | Concentration response (g / ha) | | | | | | | |
|---|---|---|---|---|---|---|---|---|
| | n | Slope ± SE | LC$_{50}$ (95%CI) | Probit Trend | | | | RR$_{50}$ (95%CI) |
| | | | | Test for Slope | | Test for GoF | | |
| | | | | $X^2$ | $P > X^2$ | $X^2$ | $P > X^2$ | |
| Year 2016 | | | | | | | | |
| 0-D | 960 | 0.758 ± 0.104 | 91.23 (69.39–119.6) | 53.710 | < .0001 | 1.8460 | 0.0004 | 1 |
| 2-D | 960 | 0.667 ± 0.158 | 328.9 (162.10–520.2) | 17.871 | < .0001 | 2.2846 | < .0001 | 2.67 (1.56–4.56) |
| 4-D | 960 | 0.445 ± 0.056 | 261.7 (161.60–520.9) | 64.162 | < .0001 | 1.5275 | 0.0121 | 3.04 (1.65–5.61) |
| 7-D | 720 | 0.287 ± 0.061 | 844.8 (339.26–5201.6) | 22.183 | < .0001 | 1.4790 | 0.0356 | 9.81 (2.94–32.77) |
| 9-D | 720 | 0.357 ± 0.065 | 1763.3 (483.60–36507) | 30.04 | < .0001 | 1.4000 | 0.0608 | 20.48 (3.46–121.12) |
| Year 2017 | | | | | | | | |
| 0-D | 960 | 0.254 ± 0.070 | 156.9 (52.31–2183) | 13.034 | 0.0003 | 5.3977 | < .0001 | 1 |
| 2-D | 960 | 0.110 ± 0.082 | 2284306 (-) | 1.801 | 0.1796 | 3.6038 | < .0001 | 10744 (0.001–1043569415) |
| 4-D | 960 | 0.290 ± 0.057 | 1287 (406.8–15201) | 25.803 | < .0001 | 1.8681 | 0.0003 | 8.95 (1.14–70.29) |
| 7-D | 960 | 0.199 ± 0.049 | 62473 (4961–77476623) | 16.342 | < .0001 | 0.8597 | 0.7378 | 434.53 (7.72–24472.75) |

LC$_{50}$ values were calculated in g / ha

Mortality was scored at 7 Days After Exposure

Differences among RR$_{50}$ values are significant if 95% CI do not 1nclude 1.0

RR50 and 95% CI calculated using formula from Robertson et al. [39]

and the rate of changes in mortality as residues dissipate [43]. Our results demonstrated that the high concentration of all insecticides had the ability to remain present (ppm) till 11-DPT (Fig 5); however, no insecticide produced 100% cumulative mortality by contact even using a susceptible laboratory colony that has been successively reared for over 20 years with no infusion of field population (Figs 1–4). The organophosphate (acephate) showed greater toxicity than any other insecticide tested on *L. lineolaris* adults, with initial mortality of >50% for 2016 and > 30% for 2017 at 0-DAE on leaves pulled at 0-DPT for the high concentration. Mortality rapidly increased to 81.7 ± 23.4 and 63.3 ± 28.8 (SE) 1-day after exposure (DAE) for 2016 and 2017, respectively attaining 94.5 ± 9.5 and 95.4 ± 7.6 6-DAE for each year. This high residual activity persisted to 4-DPT with mortality >90% decreasing to >80% after 7-DPT in 2016. However, it is important to clarify, that the high mortality observed during this study is not likely to occur in field populations of *L. lineolaris* due to the current levels of resistance to acephate previously reported for this insect throughout the Mississippi Delta cropping region [1, 6–13]. For example, an earlier study demonstrated that acephate plus bifenthrin (a pyrethroid), bifenthrin alone, sulfoxaflor, dicrotophos (an organophosphate), and thiamethoxam (a neonicotinoid) all provided better control than acephate alone [44]. However, those results differed from a more current study [45], where sulfoxaflor, acephate, and dicrotophos reduce larger populations of *L. lineolaris* in cotton plots producing higher yields, respectively. Yet, acephate, novaluron, and flonicamid showed the longest residual activity with higher concentrations in parts per billion in cotton leaf tissue [45].

Persistence is one of the main characteristics of residual activity, described in half-life, which is a comparative measure of the time needed for the chemical to degrade [46]. Therefore, the longer the insecticide's half-life, the slower the degradation, which likely translate to increased persistence of the insecticide. In some cases, persistent insecticide residues are desirable due to its long-term control and the reduction in the need for multiple applications of the insecticide in high pressure situations. Our results demonstrated that the pyrethroid lambda-cyhalothrin could be as persistent insecticide as acephate. Interestingly acephate's insecticidal residues (114.27 ppm) were >13.3 –fold higher than the pyrethroid (lambda cyhalothrin) (8.84 ppm) (Fig 5), yet the cumulative mortality attained with pyrethroid was almost as high as it was with acephate for both years (Figs 1 and 2). However, the initial mortality with acephate was 5-fold higher than lambda-cyhalothrin on the leaf sample pulled at 0-DPT in 2016. Similar initial mortality was observed between both insecticides in 2017, and cumulative mortalities were comparable for both insecticides at all evaluation periods; although, it increased faster with acephate than with lambda-cyhalothrin. This could explain why acephate is the most desirable control option to cotton growers, not just for the visible and faster control but for the reduction in needed insecticide applications [45]. Conversely, our results corroborated with Palmquist' et al. 2012 [46] findings, which reported that pyrethroid insecticides as a group replaced many organophosphate insecticides due to better selectivity on target pest and less persistence than organochlorine insecticides. William et al. 2003 [43] demonstrated that the survival of female *Anaphes iole* (Hymenoptera: Mymiridae) was four times longer when exposed to the pyrethroid lambda-cyhalothrin than to the organophosphate acephate or the nicotinoid imidacloprid.

Imidacloprid, which is the world's leading insecticide, was released in 2003 and has been approved for controlling infectious disease vectors [47]. Since then, it has been successfully used as a systemic and contact insecticide to control *L. lineolaris*, alone or in combination with other insecticides [3]. *L. lineolaris'* resistance to this insecticide it was first reported for the first time in 2012 [12], which was subsequently followed by several additional papers demonstrating the low response of this insect to neonicotinoids [13, 15, 33, 48, 49]. Our results confirmed their findings; however, our outcomes cannot be referred as a resistance, since our

observations were limited to a susceptible laboratory colony established in 1998 with not infusion from field population [15]. The response of *L. lineolaris* to imidacloprid and its residual activity found in our study was significantly lower compared to acephate and lambda-cyhalothrin during 2016 and 2017 (Figs 1–3). Although its insecticidal residues for the high concentration (8.35 ppm) on leaf samples collected at 0-DPT was as high as the pyrethroid insecticide (8.84 ppm) (Fig 5), its cumulative mortality did not exceed 40% and its systematic effect was delayed; indicating the effectiveness of contact insecticides over those systemics in nature [11]. The low cumulative mortality attained after 2-DPT period is supported by the low insecticidal residues found on the leaf samples tested. The ppm values decreased 6.62-fold from 0-DPT (8.35 ppm) to 2-DPT (1.27 ppm) and 596-fold to 9-DPT (0.014 ppm). These results are comparable to a recent study, that evaluated eight insecticides including imidacloprid, finding that this neonicotinoid had the lowest efficacy against *L. lineolaris* in cotton [49]. They reported that all insecticide treatments significantly reduced the *L. lineolaris* nymph populations; mortality due to imidacloprid, however, did not differ among untreated controls 14 days after treatment. Additionally, they noted that all insecticides treatments resulted in significantly higher yields than the untreated control, but imidacloprid treated plots had the lowest yield among treatments. Likewise, Graham and Smith, 2021 [50] corroborated with our finding, where they reported no significant differences among all treatments (insecticides and untreated control) seven days after treatment, where the total populations (nymph and adult) treated with imidacloprid were larger than untreated control.

Similar to imidacloprid, sulfoxaflor is a systemic insecticide that incurs its toxicity through contact and oral ingestion [51]. Therefore, based on *L. lineolaris* food consumption their systemic effect could be equal on nymphs and adults; however, the contact action could be higher in nymphs because they are less mobile than adults [15]. In this study, the mobility of adults was restrained, where their arena was a folded-treated cotton leaf in a 30 mL cup. Therefore, adults were forced to be in contact and feed on the cotton treated leaf. Our results demonstrated that efficacy of sulfoxaflor on *L. lineolaris* adults was highly superior to imidacloprid. The cumulative mortality trend for leaves pulled at 0-DPT was comparable to the pyrethroid lambda-cyhalothrin (Figs 3 and 4). Yet, the residual effect was as low as imidacloprid for leaf samples collected at 4, 7, and 9-DPT. The low cumulative mortality after the 4-DPT was unexpected, since the insecticidal residues test for sulfoxaflor was 2.18- and 3.82-fold higher than lambda-cyhalothrin for leaves pulled at 4-, and 7-DPT, respectively. These results deferred from other field studies using eight insecticides including sulfoxaflor, which demonstrated that sulfoxaflor's efficacy against *L. lineolaris* in cotton was higher than any other insecticide, however they did not differ when compared to novaluron (an effective insect growth regulator) or acephate [49]. The authors reported no significant differences among 4, 7, and 14 days after treatment, suggesting that sulfoxaflor's insecticide residual continued to 14 days after application. No differences in residual effect were found between sulfoxaflor and acephate [49]. Comparably, Siebert et al. 2012 [52] demonstrated similar residual efficacy levels at evaluation intervals >6 days after application between sulfoxaflor and acephate against moderately tolerant populations of *L. lineolaris* in 12 mid-southern U.S. locations from 2008 through 2010. Although, both studies were conducted a decade a part, acephate continually proves to demonstrate highly efficacious for control of *L. lineolaris* relative to other classes of insecticides regardless of any inherent resistance development.

Today, sulfoxaflor is considered one of the most promising insecticides used to control *L. lineolaris*. Our results however, indicated that the organophosphate acephate and the pyrethroid lambda-cyhalothrin exhibited higher mortality and longer residual activity than the sulfoxaflor (a sulfoxamine) on treated leaves with the highest concentration. *L. lineolaris* adults are significantly more susceptible to contact than systemic insecticides and due to its residual

effect, acephate could kill over 80% of the TPB population 7-DPT. Conversely, in agricultural situations, selection of insecticides for *L. lineolaris* control needs to be closely related to dynamic resistance levels while preserving natural enemies of insect pests and pollinators, both of which can directly increase crop yields. On the other hand, environmental conditions could play and import roll on the insect residual activity.

## Supporting information

**S1 File. Probit by day by insecticide 2016–2017.**
(XLSX)

## Acknowledgments

The authors would like to thank Tabatha Nelson, Arnell Patterson (RIP), and Henry Winter ARS-USDA, Southern Insect Management Research Unit (SIMRU), Stoneville, for their assistance with insect rearing, laboratory assays and field leave collection. Owen Houston, Phil Powell, G. (USDA ARS Southern Insect Management Research Unit, Stoneville, MS) for their assistance with the field plots preparation. To Michael Huoni, student trainee RIMRU for his valuable help in data entry. Mention of trade names or commercial products in this publication is solely for the purpose of providing specific information and does not imply recommendation or endorsement by the U.S. Department of Agriculture or the Agricultural Research Service.

## Author Contributions

**Conceptualization:** Maribel Portilla.

**Data curation:** Maribel Portilla.

**Formal analysis:** Maribel Portilla.

**Investigation:** Maribel Portilla, Nathan Little, Clint Allen, Yu Cheng Zhu.

**Methodology:** Maribel Portilla, Nathan Little, Clint Allen, Yu Cheng Zhu.

**Supervision:** Maribel Portilla.

**Validation:** Maribel Portilla.

**Visualization:** Maribel Portilla.

**Writing – original draft:** Maribel Portilla.

**Writing – review & editing:** Nathan Little, Clint Allen, Yu Cheng Zhu.

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
