## [Decision Letter · Decision Letter 0]

31 Jan 2024

PONE-D-23-26095

Effectiveness and residual activity of four common insecticides used in the Mississippi Delta to control tarnished plant bugs in cotton

PLOS ONE

Dear Dr. Portilla,

Thank you for submitting your manuscript to PLOS ONE. After careful consideration, we feel that it has merit but does not fully meet PLOS ONE’s publication criteria as it currently stands. Therefore, we invite you to submit a revised version of the manuscript that addresses the points raised during the review process.

**Dr Portilla**

The manuscript is well written and deserve publication on PlosOne Journal. I'm confident that reviewers comments will improve more the quality of that paper.

Specific comment: Be more accurate when summarizing your findings usefulness at conclusion section.

Good luck

We look forward to receiving your revised manuscript.

Kind regards,

Rachid Bouharroud

Academic Editor

PLOS ONE

Journal Requirements:

Reviewers' comments:

Reviewer's Responses to Questions

**Comments to the Author**

1. Is the manuscript technically sound, and do the data support the conclusions?

Reviewer #1: Yes

Reviewer #2: Yes

2. Has the statistical analysis been performed appropriately and rigorously? 

Reviewer #1: Yes

Reviewer #2: Yes

3. Have the authors made all data underlying the findings in their manuscript fully available?

Reviewer #1: Yes

Reviewer #2: No

4. Is the manuscript presented in an intelligible fashion and written in standard English?

Reviewer #1: Yes

Reviewer #2: Yes

5. Review Comments to the Author

Reviewer #1: 134: How were the doses determined? These seems high for 2023 so extremely high for 2016.

164: Instead of 0 DPT, I suggest replacing with at each time interval

167: TPB were not given any food source for seven days? In 463, states they were required to feed on the leaf knowing they prefer squares.

168: reads like all three concentrations were placed in the same cup instead of individual cups

206: The placement for table 2 is weird since it includes all the insecticides, however only acephate has been discussed at this point in the paper.

344: Depending on when the rainfall occurred after application, rainfall may not have had a great of an impact on results as suggested in the text.

384: Rainfall can and does affect insecticide but when did the rainfall occur and how much. Important to go into this a bit more.

405: bifenthrin is spelled incorrectly

407: I would disagree that acephate and dicrotophos are the two most used insecticides. Acephate plus bifenthrin, transform, and novaluron are the most commonly used products.

I am not sure how this paper is relative to today and the TPB management. What was found in this paper is not what is reflected. Transform is by far the best TPB product for midsouth control however, this paper would not suggest that. It’s important to take the data in the paper and using some field efficacy studies to correlate what was discovered here and what is occurring in the field. 2016 to 2023 is a large time gap, things have change. Some of this was done in the conclusion.

All the tables and figures at the end are very unclear. All of these need to be updated.

Reviewer #2: Hello,

The authors should reread the article to correct some words and improve the quality of the study.

Alternatively, the authors should include more information that clarifies materials and methods, especially in “Field plots and applications” that’s the weather data of the field (humidity, temperature, precipitation…) it needs to be added. Also, it’s preferred to give the coordinates of the study area (altitude, longitude).

For the scientific name of insect “Lygus lineolaris”, it should be abbreviated the genus Lygus by using the letters L. in all paragraphs.

Please, some references like 10,20,21,22,28 ,29,30,31,32,33 and 51 are missing in the article, the authors require to include them. In addition, in the line 86 they are two numbers (102 and 106) are mentioned as references but in references listed, they are no more than 52.

6. PLOS authors have the option to publish the peer review history of their article (what does this mean?). If published, this will include your full peer review and any attached files.

Reviewer #1: No

Reviewer #2: **Yes: **Nezha Ait Taadaouit

---

## [Author Response · Author response to Decision Letter 0]

28 Mar 2024

Reviewer #1: 

134: How were the doses determined? These seems high for 2023 so extremely high for 2016.

Response: Dose responses for insects to bioassays are known to be logarithmic in nature, and we took this into account when selecting doses to elicit efficacy and residual activity of the insecticides sprayed in the field for this study. For safety and practicality, we limited the study to three doses. Treatment rates used reflected the highest labeled rate (High) for a given insecticide for TPB control in the U.S. in 2016, 1/10th (Medium), and 100th (Low) of the high rate. 

164: Instead of 0 DPT, I suggest replacing with at each time interval.

Response: Replaced and restructured. Lines 165-167 and line 171.

167: TPB were not given any food source for seven days? In 463, states they were required to feed on the leaf knowing they prefer squares.

Response: In the MS lines 168-169 was stated that “A 2-d old L. lineolaris adult (unknown sex) was released to each cup that contained the treated leaf with the three concentrations of each tested insecticide”. So, insects were fed with cotton treated leaves. Yes, TPB prefers squares, however being a sucking insect, adults and nymphs can feed on any part of the plant. In our case, the area on which the insect supposed to be exposed and fed without source food replacement was important. A single folded leaf last 7 days to evaluate the residual. 

168: reads like all three concentrations were placed in the same cup instead of individual cups.

Response: Agreed. The sentence was shortened and now is written as: “A 2-d old L. lineolaris adult (unknown sex) was released to each cup that contained the treated leaf”. Now lines 168-169.

206: The placement for table 2 is weird since it includes all the insecticides, however only acephate has been discussed at this point in the paper.

Response: Based on the guideline of the journal the tables supposed to be placed immediately after the paragraph that it has been mentioned. 

344: Depending on when the rainfall occurred after application, rainfall may not have had a great of an impact on results as suggested in the text.

Response: It was not our intention to focus on rainfall; however, because the rain that occurred during July – August 2017, the third and fourth spray was not possible until late August and early September. Information about the precipitation days after treatment for the replicates 2, 3, and 4 was included in results for clarification, now lines 207-212, also some clarification was included in table 1 (lines 140-141). 

384: Rainfall can and does affect insecticide but when did the rainfall occur and how much. Important to go into this a bit more.

Response: Information on when the rainfall occurred and how much was added to the text (lines 140-142 and 207-2012).

405: bifenthrin is spelled incorrectly

Response: Corrected

407: I would disagree that acephate and dicrotophos are the two most used insecticides. Acephate plus bifenthrin, transform, and novaluron are the most commonly used products.

Response: Sentence was restructured as follow (lines 426-430): “those results differed from a more current study [45], where sulfoxaflor, acephate, and dicrotophos reduce larger populations of L. lineolaris in cotton plots producing higher yields, respectively. Yet, acephate, novaluron, and flonicamid showed the longest residual activity with higher concentrations in parts per billion in cotton leaf tissue [46].”

I am not sure how this paper is relative to today and the TPB management. What was found in this paper is not what is reflected. Transform is by far the best TPB product for midsouth control however, this paper would not suggest that. It’s important to take the data in the paper and using some field efficacy studies to correlate what was discovered here and what is occurring in the field. 2016 to 2023 is a large time gap, things have change. Some of this was done in the conclusion.

Response: We agreed with the reviewer, it is a large time gap. However, the information presented in this investigation demonstrate that Acephate continued to be the most effective insecticides against tarnished plan bug and its residual effects last longer than any other insecticide available for TBP control, as was demonstrated in a study conducted in 2023 cited in discussion, lines 426 and 430. 

All the tables and figures at the end are very unclear. All of these need to be updated.

Response: Tables are presented like any other published PROBIT analysis table. About the graphs, we believe that scattered lines are be the best way to present cumulative mortality data dose response. Place one graph below another show the sequency of the residual activity days after treatment. And it is clearly comparable between years. Therefore, very respectfully no changes were made. However, additional information was added to the title of the figures for clarity. If the reviewer has any suggestions for improving tables and figures we will be more than happy to address and make any changes. 

Reviewer #2: 

Hello, The authors should reread the article to correct some words and improve the quality of the study.

Alternatively, the authors should include more information that clarifies materials and methods, especially in “Field plots and applications” that’s the weather data of the field (humidity, temperature, precipitation…) it needs to be added. Also, it’s preferred to give the coordinates of the study area (altitude, longitude).

Response: Field plots and applications are well described in full and half page (lines 106-133) including a table (1) (page 8) with doses per each insecticide and dates. However, altitude and longitude were added now line 109. Some information on precipitation was added to results now lines 207-212, discussion lines 399-407, also some clarification was included in table 1 (lines 140-141). 

For the scientific name of insect “Lygus lineolaris”, it should be abbreviated the genus Lygus by using the letters L. in all paragraphs.

Response: Corrected throughout the manuscript 

Please, some references like 10,20,21,22,28 ,29,30,31,32,33 and 51 are missing in the article, the authors require to include them. In addition, in the line 86 they are two numbers (102 and 106) are mentioned as references but in references listed, they are no more than 52.

Response: reference 10 is cited in line 60 noted as [9-15]; references 20,21,22 are cited in line 69 noted as [19-23], references 28, 29, 30,31, 32, 33 are cited in line 84 noted as [27-34]; reference 51 was missing and was added now line 463. References 102 and 106 were miss-typed and corrected for 12 and 16, line 8

---

## [Editor Report · Decision Letter 1]

2 Apr 2024

Effectiveness and residual activity of four common insecticides used in the Mississippi Delta to control tarnished plant bugs in cotton

PONE-D-23-26095R1

Dear Dr. Portilla,

We’re pleased to inform you that your manuscript has been judged scientifically suitable for publication and will be formally accepted for publication once it meets all outstanding technical requirements.

Kind regards,

Rachid Bouharroud

Academic Editor

PLOS ONE